# 'Optimising PharmacoTherapy In the multimorbid elderly in primary CAre' (OPTICA) to improve medication appropriateness: study protocol of a cluster randomised controlled trial

Katharina Tabea Jungo,[1] Zsofia Rozsnyai,[1] Sophie Mantelli,[1] Carmen Floriani,[1] Axel Lennart Löwe,[1,2] Fanny Lindemann,[1] Nathalie Schwab,[1,2] Rahel Meier,[3] Lamia Elloumi,[4,5] Corlina Johanna Alida Huibers,[6] Bastiaan Theodoor Gerard Marie Sallevelt,[7] Michiel C Meulendijk,[8] Emily Reeve,[9,10,11] Martin Feller,[1,2] Claudio Schneider,[2] Heinz Bhend,[12] Pius M Bürki,[13] S Trelle,[14] Marco Spruit,[4] Matthias Schwenkglenks,[15,16] Nicolas Rodondi,[1,2] Sven Streit[1]

For numbered affiliations see end of article.

**Correspondence to**
Professor Sven Streit;
sven.streit@biham.unibe.ch

## ABSTRACT

**Introduction** Multimorbidity and polypharmacy are major risk factors for potentially inappropriate prescribing (eg, overprescribing and underprescribing), and systematic medication reviews are complex and time consuming. In this trial, the investigators aim to determine if a systematic software-based medication review improves medication appropriateness more than standard care in older, multimorbid patients with polypharmacy.

**Methods and analysis** Optimising PharmacoTherapy In the multimorbid elderly in primary CAre is a cluster randomised controlled trial that will include outpatients from the Swiss primary care setting, aged ≥65 years with ≥three chronic medical conditions and concurrent use of ≥five chronic medications. Patients treated by the same general practitioner (GP) constitute a cluster, and clusters are randomised 1:1 to either a standard care sham intervention, in which the GP discusses with the patient if the medication list is complete, or a systematic medication review intervention based on the use of the 'Systematic Tool to Reduce Inappropriate Prescribing'-Assistant (STRIPA). STRIPA is a web-based clinical decision support system that helps customise medication reviews. It is based on the validated 'Screening Tool of Older Person's Prescriptions' (STOPP) and 'Screening Tool to Alert doctors to Right Treatment' (START) criteria to detect potentially inappropriate prescribing. The trial's follow-up period is 12 months. Outcomes will be assessed at baseline, 6 and 12 months. The primary endpoint is medication appropriateness, as measured jointly by the change in the Medication Appropriateness Index (MAI) and Assessment of Underutilisation (AOU). Secondary endpoints include the degree of polypharmacy, overprescribing and underprescribing, the number of falls and fractures, quality of life, the amount of formal and informal care received by patients, survival, patients' quality adjusted life years, patients' medical costs, cost-effectiveness of the intervention, percentage of recommendations accepted by

## Strengths and limitations of this study

► The Optimising PharmacoTherapy In the multimorbid elderly in primary CAre (OPTICA) trial is the first randomised controlled trial to examine the effect of the 'Systematic Tool to Reduce Inappropriate Prescribing'-Assistant, a software-assisted clinical decision support tool, on medication appropriateness in older, multimorbid patients with polypharmacy in a primary care setting.

► OPTICA is the first randomised controlled trial to test the use of software-based structured medication reviews in Swiss primary care.

► The OPTICA trial demonstrates how linked and coded data from electronic medical records can be used to evaluate primary care interventions in a randomised controlled trial setting. The investigators limit selection bias by using screening lists with a random sample of potentially eligible patients and randomising general practitioners (GPs) after patient recruitment is complete, but cannot eliminate the risk of selection bias.

► The investigators chose a sham intervention in accordance with usual care in the control group to improve patient blinding but, by design, they could not blind GPs.

GPs, percentage of recommendation rejected by GPs and patients' willingness to have medications deprescribed.
**Ethics and dissemination** The ethics committee of the canton of Bern in Switzerland approved the trial protocol. The results of this trial will be published in a peer-reviewed journal.

**Main funding** Swiss National Science Foundation, National Research Programme (NRP 74) 'Smarter Healthcare'.

**Trial registration numbers** Clinicaltrials.gov (NCT03724539), KOFAM (Swiss national portal) (SNCTP000003060), Universal Trial Number (U1111-1226-8013).

## INTRODUCTION

Globally, there is a high prevalence of multimorbidity and polypharmacy in people more than 65 years old.[1 2] Multimorbidity is commonly defined as the coexistence of three or more chronic diseases,[3] and polypharmacy is commonly defined as the regular intake of five or more medications.[4] Polypharmacy is often caused by multimorbidity and is linked to a high risk of potentially inappropriate prescribing,[5 6] which has three main elements, namely: (1) overuse, (2) underuse and (3) inappropriate use of medications (ie, wrong dose/medication for the indication).[7]

Appropriate polypharmacy denotes a situation in which 'medication use is optimised according to the patients' clinical needs' and in which patients 'receive the most appropriate combinations of medications based on the best available evidence'.[8] While appropriate polypharmacy can improve quality of life and prevent consequences of disease, inappropriate polypharmacy can harm patients' health.[4] For instance, it can increase the risk of falls and fractures,[9] lead to cognitive decline,[10 11] and it can reduce quality of life.[9] Polypharmacy also increases the risk of drug–drug interactions, drug–disease interactions and adverse drug events.[12–15] Treatment of older patients with multimorbidity and polypharmacy is a complex problem in primary care and other medical fields and both conditions are increasingly common as populations age. Due to the relatively small number of randomised controlled trials on different interventions for the management of multimorbid people, there currently remain uncertainties about the effectiveness of these interventions.[16] Medication reviews in older adults with multimorbidity and polypharmacy can be complicated and time-consuming.[17] General practitioners (GPs) have reported that time limitations and lack of user-friendly and reliable tools pose significant barriers to regular medication optimisation activities in practice.[18]

Current evidence is ambiguous on whether review interventions that aim at optimising medication can reduce inappropriate polypharmacy. A systematic review and meta-analysis of short-term medication review interventions published in 2017 showed that such isolated medication reviews have an impact on drug-related outcomes, but only minimally influence clinical outcomes and have no impact on quality of life.[19] A more recent systematic review and meta-analysis on interventions to improve the appropriateness of medication use in older people showed that such interventions may be beneficial for reducing potential prescribing omissions, but that it remains uncertain whether they improve the appropriateness of medication use.[20]

The increasing use of electronic medical records (EMR) and electronic prescribing has opened opportunities to incorporate web-based clinical decision support systems (CDSS) at the point of care. For instance, a complex intervention with an informatics tool and financial incentives has led to a reduction in high-risk prescribing of non-steroidal anti-inflammatory drugs and antiplatelet medications.[21] Medication optimisation studies that use CDSS enabled by electronic medical records have shown that the interventions are feasible and acceptable to clinicians, but evidence that they lead to more appropriate use of medications in general and improve clinical outcomes is still limited.[22]

### Objectives

The primary aim of the cluster-randomised 'Optimising PharmacoTherapy In the multimorbid elderly in primary CAre' (OPTICA) trial is to test whether the use of a systematic, software-assisted medication review intervention leads to a more appropriate use of medications than a usual care sham intervention. This primary outcome will be measured by the Medication Appropriateness Index (MAI) and the Assessment of Underutilisation (AOU). Secondary outcomes are the degree of polypharmacy, the degree of overprescribing, the degree of underprescribing, the number of falls and fractures, quality of life, the amount of formal and informal care received by patients, survival, patients' quality adjusted life years (QALYs), patients' medical costs, cost-effectiveness of the intervention, percentage of recommendations accepted and rejected by GPs, and patients' willingness to have medications deprescribed.

## METHODS AND ANALYSIS
### General study design and setting

The OPTICA trial is a cluster randomised controlled trial (RCT), coordinated at the Institute of Primary Healthcare of the University of Bern (BIHAM). Participating GPs, who will each systematically recruit multimorbid, older patients with polypharmacy, define the clusters. Through randomisation, the GPs will be allocated to the structured medication review or a usual care based sham intervention.

The investigators used the Standard Protocol Items: Recommendations for Interventional Trials (SPIRIT) checklist when they wrote this protocol.[23] Please refer to figure 1 for the study flow chart and figure 2 for a description of the data flow within the OPTICA trial.

### Cluster definition

The trial will be conducted in about 40 primary care offices in the German-speaking regions of Switzerland. Each participating GP, who prescribes the medication of his/her patients, constitutes a cluster. GPs were recruited as subinvestigators from October 2017 until June 2018 from the pool of teaching physicians at the University of Bern and from the group of GPs who attended project presentations hosted by the BIHAM throughout 2017 and 2018. Additional GPs will be recruited throughout 2019

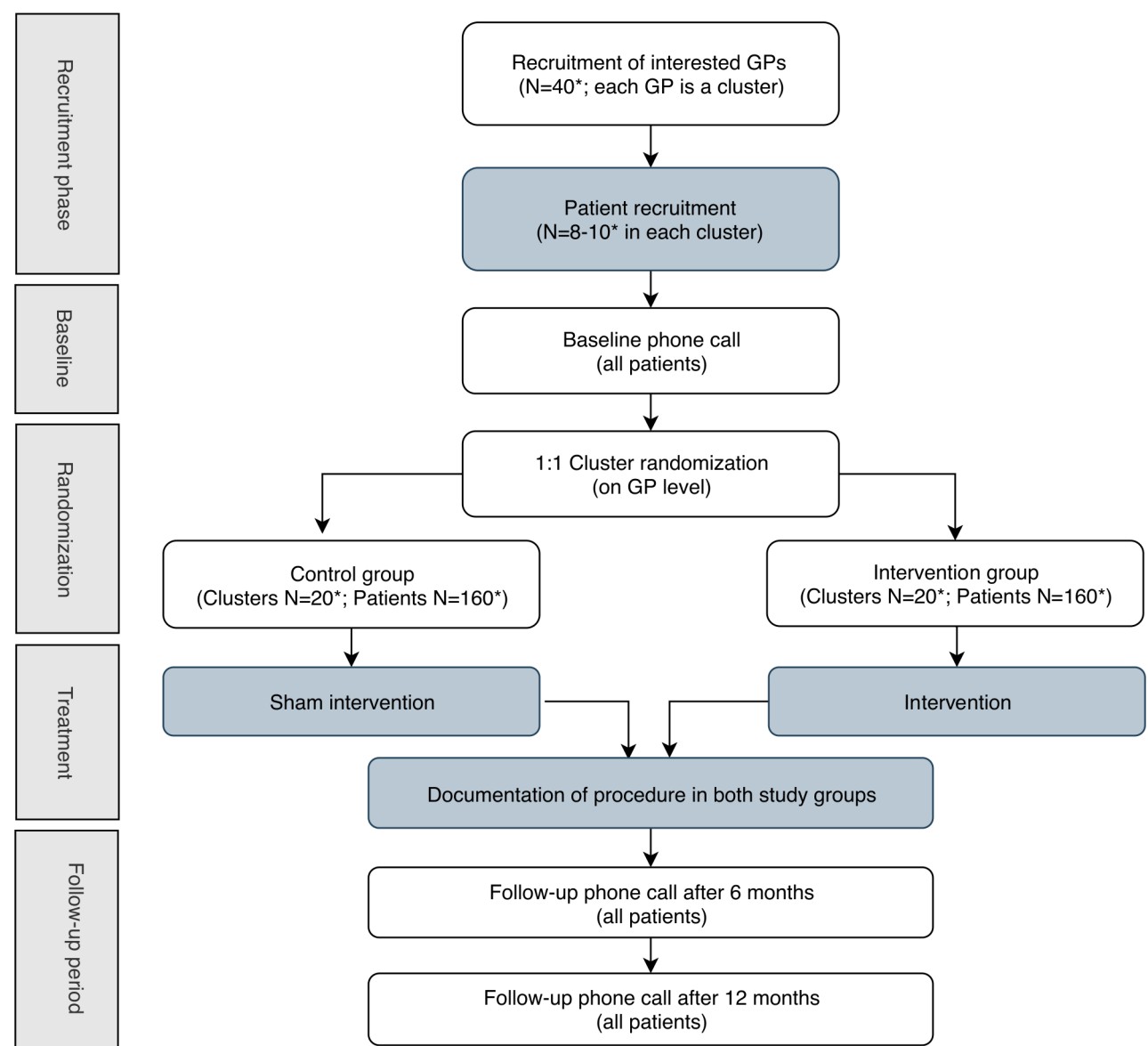

**Figure 1** OPTICA trial flow chart. Flow diagram of the progress through the phases of the OPTICA trial. In dark grey: steps done by general practitioners. In white: steps done by study team at BIHAM. *Target number. BIHAM, Institute of Primary Healthcare of the University of Bern; GPs, general practitioners; OPTICA, Optimising PharmacoTherapy In the multimorbid elderly in primary CAre; STRIPA, Systematic Tool to Reduce Inappropriate Prescribing-Assistant.

to replace GPs who had to withdraw from the study, as the data export to the FIRE database did not work in their GP office. Engagement with GPs started early in anticipation of slow recruitment, which has been reported in previous studies.[24 25] All eligible GPs who showed interest in the OPTICA trial were visited by the investigators at their GP office and given a detailed explanation of the trial.[26] As of July 2019, 83% of the participating GPs are male. Eighty per cent of the GPs work in group practice, while 20% work in single practice. Twenty-nine per cent of GPs work in the countryside and 71% work in urban and suburban areas.

### Randomisation
Randomisation is done after the cluster has been completed. The randomisation is done centrally in a

web-based system (REDCap) by a study team member after all cluster information has been entered. Each participating GP is allocated 1:1 to the intervention group or the control group, using unstratified randomisation with a random sequence of block sizes of two and four. An independent statistician, who is otherwise not involved in the trial, generated the randomisation list. To uphold the concealment of allocation only system administrators who are otherwise not involved in the trial can access the randomisation list.

### Eligibility criteria
#### Inclusion criteria
##### GP level
To be eligible for participation, GPs must be participating in the 'Family medicine ICPC Research using Electronic

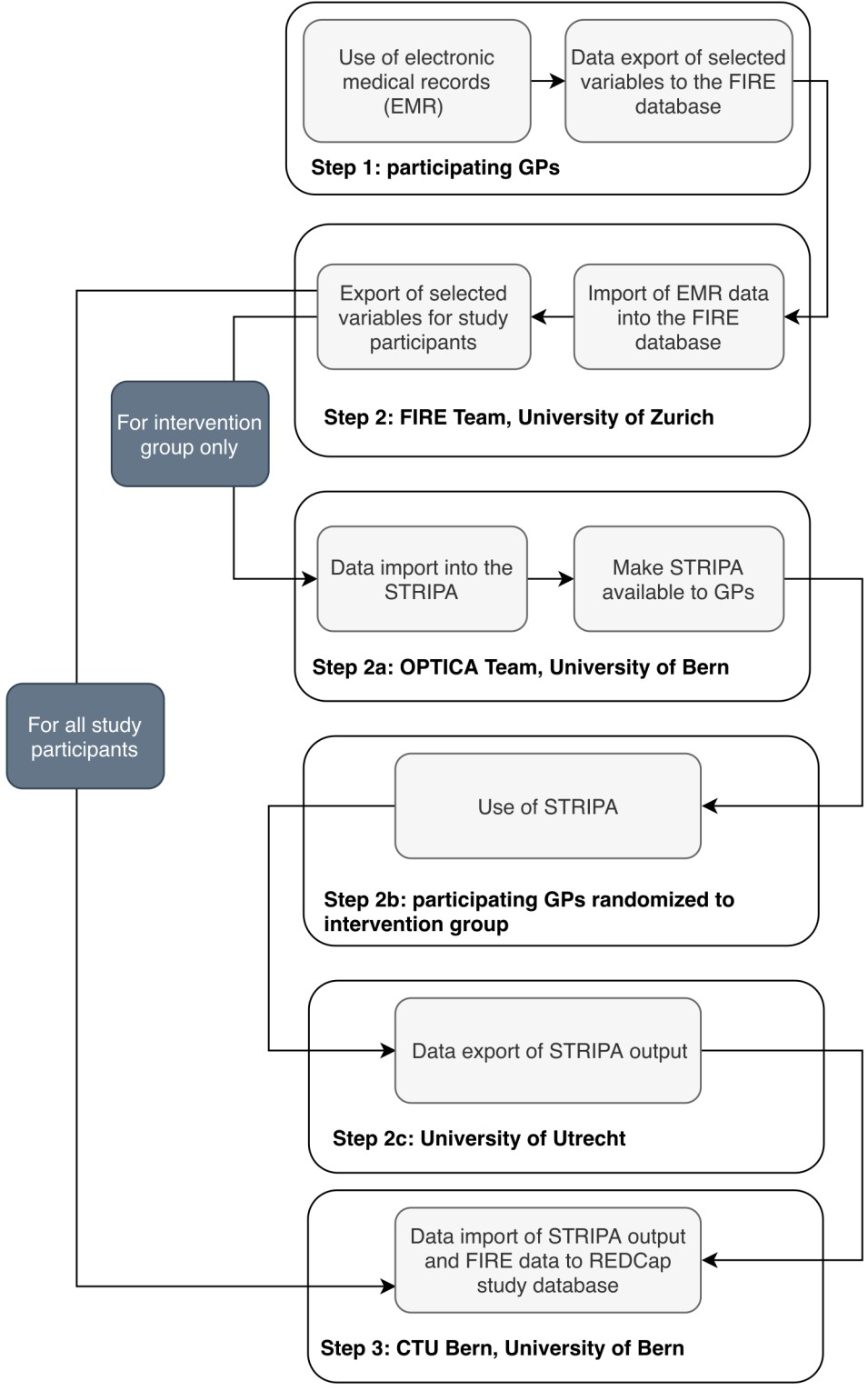

**Figure 2** OPTICA data flow chart. Step-by-step explanation of the data flow during the OPTICA trial. EMR, electronic medical record; FIRE, Family medicine ICPC Research using Electronic medical records; GPs, general practitioners; OPTICA, Optimising PharmacoTherapy In the multimorbid elderly in primary CAre; STRIPA, Systematic Tool to Reduce Inappropriate Prescribing-Assistant.

medical records' (FIRE) project led by the Institute of Primary Care of the University of Zurich.[27] The EMR they use in their GP office have to be compatible with the FIRE database, which is a database with anonymous data from the EMR of more than 500 GPs in Switzerland. The database of the FIRE project contains administrative data, vital data, laboratory values, International Classification of Primary Care 2 (ICPC-2) codes for diagnoses and information on medications prescribed. Participating GPs can, only for the purpose of this trial, identify individual

patients in the usually anonymous FIRE database, so that their data can be used in this trial. All participating GPs must complete an online training for Good Clinical Practice (GCP).

### Patient level

Patients must be enlisted by one of the participating GPs, so that this GP is their main prescribing physician and they must see their GP regularly. The investigators left the definition of regularity to the judgement of participating physicians. Patients must be ≥65 years of age, have ≥three chronic diseases based on ICPC-2 coding or based on the GP's clinical decision (multimorbidity), and regularly take ≥five medications (polypharmacy). Written informed consent must be sought from patients or, if patients suffer from cognitive impairment, their relative prior to enrolment. This consent includes the right to obtain data from the FIRE project about the study participant.

### Exclusion criteria
### GP level

In group practices, only one GP can take part in the trial.

### Patient level

To maximise the generalisability of the study population, the investigators kept exclusion criteria to a minimum. They excluded patients currently taking part in another interventional study.

### Intervention

The intervention, a systematic medication review, includes the use of a web-based CDSS called 'Systematic Tool to Reduce Inappropriate Prescribing'-Assistant (STRIPA), developed by a team from the Utrecht University and the University Medical Centre Utrecht in the Netherlands.[28] STRIPA is based on all algorithms of the 'Screening Tool of Older Person's Prescriptions' (STOPP) and 'Screening Tool to Alert doctors to Right Treatment' (START) criteria V.2,[29] which are expert-consensus lists of inappropriate and appropriate medications for older adults that consider coexisting medical conditions and are thus suited for optimising the prescriptions of multimorbid older patients.[30] STRIPA combines implicit and explicit prescribing tools and takes a stepped approach that actively encourages patient involvement in decision-making.[28 31] Taking into account medications, diagnoses, laboratory values and vital data, the STRIPA generates recommendations for physicians about 'underprescribing, ineffective prescribing, overprescribing, side effects, contraindications, (…) drug–drug and drug–disease interactions, incorrect dosages/dosing frequencies and practical intake issues'.[28] The intervention's recommendations allow patients and GPs to conduct a shared decision-making process about patients' medication intake, so patient preferences play an important role in this trial.

A STRIPA validation study with GPs based on two test cases showed appropriate prescribing decisions increased and inappropriate decisions decreased during

a medication review of an older, multimorbid patient with polypharmacy.[31] STRIPA is being tested in the European multicentre clinical trial 'OPtimising thERapy to prevent Avoidable hospital admissions in Multimorbid older people' (the OPERAM trial) in the hospital setting to find out if it can reduce drug-related hospital re-admissions.[32]

This study adapted the intervention from the OPERAM trial so it could be tested in the primary care setting.[33] The intervention comprises six steps:

1. This data will be imported from the FIRE database into the STRIPA: patients' baseline characteristics (eg, sex, age); vital data and laboratory measurements (eg, kidney function, blood pressure), patients' signs and symptoms, different scores (eg, HAS-BLED score for major bleeding risk), ICPC-2 coded diagnoses, and medications entered as Anatomical Therapeutic Chemical codes.
2. GPs log in to STRIPA and see a list with their recruited patients' identifiers. When they click on a patient's ID, the information exported from the EMR to STRIPA via FIRE becomes visible (from Step 1). The GP can add unrecorded values or adjust values (eg, when more recent laboratory values are available). This might be necessary because it might take some time to obtain the FIRE data and to import it to STRIPA and hence the information might change in the meantime.
3. GPs use a drag and drop function to link each medication to a diagnosis.
4. GPs run the analysis in STRIPA and then look at STRIPA's recommendations. After the analysis is finished, STRIPA records the recommendations it generates so that they will be available for later analysis and generates a PDF report that can be saved by GPs.
5. GPs then must decide if they agree with STRIPA's advice given and if they will present it to their patient. They can follow, partly follow, or decline STRIPA's recommendations.
6. At the next appointment with the same patient, GPs will present the recommendations that they consider appropriate. GP and patient will decide together, in a shared decision-making process, which recommendations to implement. GPs will work from a checklist that explains how to conduct the shared decision-making process so that the patients' preferences are met. The shared-decision making process is based on Elwyn et al's model[34] and was adapted from the OPERAM trial. The four key elements of this shared decision-making process are 'choice talk', 'option talk', 'preference talk', and 'decision talk'.[34] Patients may follow, partly follow or decline the recommendations. Because multimorbid patients are likely to have multiple prescribers, GPs and patients can discuss recommendations with other prescribers, for example, specialists. The GP records in the REDCap study database the patient's decisions and the reasons recommendations were accepted, partially accepted, or declined. These records will be analysed later.

Before the intervention, GPs in the intervention group will watch an instruction video and read training material that will guide them through the intervention step-by-step.

### Sham intervention

Patients assigned to the control arm receive a sham intervention from their GP—a medication discussion in accordance with usual care. This means that the GP will, as usual in standard care, ask the patient if the current list of medication is complete or needs to be adapted, then share decision making with the patient about possible changes. The GPs who do the sham intervention will receive the same shared decision-making checklist as the GPs in the intervention group. The investigators explicitly ask GPs not to deviate from their usual practice when they review the patients' medication and not to use additional tools. After the sham intervention, GPs record procedures, decisions, and reasons in the REDCap study database for later analysis. This sham intervention ensures that the medication list of control group study participants, which will be used to assess study outcomes, is up-to-date, while also ensuring patients are blinded.

### Screening and enrolment

For patient recruitment, the investigators use data from the FIRE project to prepare a screening list for each GP that contains a random selection of eligible patients. On each screening list, there are about 20 patients who were randomly sampled from all enlisted patients after they met the age criteria and were determined to regularly take at least five chronic medications, based on the Pharmacy-based Cost Group model.[35–37] Since ICPC-2 coding is suboptimal in many GP offices, the investigators decided not to include the chronic disease criterion when they prepared the screening lists. From the list, GPs systematically recruit eight to 10 patients who meet the full inclusion criteria, which now include the number of chronic diseases, and do not meet any of the exclusion criteria. GPs will recruit patients starting either at the bottom of the list or the top. If GPs have finished the first screening list without recruiting the minimum amount of participants, the investigators will provide them with a second list.

Since the screening lists are compiled from a random patient sample from each GP office based on FIRE data, patients on the list may have been treated by a different GP who works in the same group practice. It is also possible that patients on the list changed their GP or have died. Since prescriptions frequently change,[38] some patients with polypharmacy might not have qualified for the screening list when data was last exported. For all these reasons, the protocol allows GPs to skip patients on the screening list if they provide an explanation, or to recruit patients who are not on the list but fulfil the inclusion and do not meet the exclusion criteria. After identifying patients on the list and verifying their eligibility, GPs then inform the patient about the study and seek their informed consent. Study-related appointments are added to the patients' regular appointments at no charge.

### Data collection

Study participants or their relatives are followed up by phone at baseline and at 6 and 12 months after study enrolment to collect data. At these times, complementary information from the FIRE database, including information about medications and chronic diseases, is imported into the REDCap study database.[27]

### Blinding procedures

The OPTICA trial is blinded to the extent the cluster design of this RCT allows. The method of partial blinding, which is similar to that used in the OPERAM trial,[32] is set out in detail in table 1.

### Follow-Up

Outcome information is collected at baseline, at 6, and at 12 months via telephone calls with patients or relatives conducted by a blinded study team member, and through FIRE database exports. The study team will make every reasonable effort to keep each patient in the study until all planned treatments and assessments have been performed. But patients may withdraw from the study or be withdrawn when they are lost to follow-up.

### Assessment of primary outcome

The primary endpoint is medication appropriateness, measured jointly by the MAI and the AOU, in each study group. While the AOU assesses underprescribing, the MAI is a tool to assess different elements of medication prescribing (eg, overprescribing, drug–drug interactions, etc).[7 39 40]

Blinded study team members assess the MAI for each regular medication study participants take and assess the AOU for each chronic condition study participants have. The investigators will use the weighted 10-item version of the MAI developed by Samsa et al.[40] However, due to the rapidly changing drug prices the MAI item on cost-effectiveness will not be included. The criteria of the MAI, including corresponding weights, can be found in table 2. Using clinical data and the predefined operational definitions for each item, the assessor rates each medication on a three-point scale ranging from A=appropriate, B=marginally appropriate, C=inappropriate. Each 'inappropriate' rating will receive the respective weight from table 2, while the weight of ratings 'appropriate' and 'marginally appropriate' will be 0. Thus, the score for each medication ranges from 0 to 17 (as the cost-effectiveness criterion will not be included).[40] A higher score indicates a greater degree of medication inappropriateness. The investigators will calculate the score for each medication and then calculate the summated score for each patient, by summing up the scores for each medication. For the AOU, assessors decide for each chronic condition if there is (i) no omission, (ii) marginal omission, or (iii) omission of indicated medication.[41–43] For each patient,

**Table 1** Blinding status and measures to assure blinding

| Role | Blinding status | How to achieve blinding |
|------|-----------------|-------------------------|
| General practitioners | Unblinded | When screening for patients and seeking informed consent, GPs are still blinded, as each cluster (GP) will be only randomised after the cluster is full (8–10 patients). The screening list GPs use for recruitment contains a random sample of potentially eligible patients, which was generated from FIRE data by a blinded study team member. All this is done to prevent selection bias. However, since GPs can recruit patients outside this list, not all study participants are recruited randomly.<br>However, the study design makes it impossible to blind GPs throughout the trial. When GPs do the medication review/discussion, they know what group they are allocated to. Nevertheless, GPs in the control arm do not know the procedures in the intervention arm, which prevents cross-contamination; if they knew the procedure in the intervention arm, they might adapt their usual care. |
| Data collectors and assessors | Blinded | The randomisation of GPs is kept concealed from the team that makes follow-up calls to avoid interviewer bias. Data collectors and assessors have no access to unblinded study information in the database or to local source data. If a SAE occurs, the study coordinator and project manager will be informed. |
| Data manager and data analyst | Unblinded | The investigators cannot blind the data managers and analysts because they can see the differences in data structure between the study groups. This is why the investigators will use a new data analyst to prepare a clean data set with truncated data to conduct a blinded analysis of the primary outcome. |
| Study coordinator and project manager, including principal investigator | Unblinded | The study coordinator, project manager and the principal investigator of the trial know the treatment allocation. They are responsible for collecting information about SAEs and performing safety assessments. |
| Patients | Partially blinded | Patients stay partially blinded. They are only given a 'high-level description' of the study question so they know that their GP has been allocated to one of two study groups, but they do not know which one. To uphold patient blinding, patients in the control group will meet their GP for a medication discussion and a shared decision-making about their prescriptions. This means patients in each study arm see GPs the same number of times during the trial and cannot guess their allocation status based on the number of consultations. |

The OPTICA trial's approach to blinding resembles the approach used in the OPERAM trial.[32]
GP, General Practitioner; OPERAM, OPtimising thERapy to prevent Avoidable hospital admissions in Multimorbid older peopl; OPTICA, Optimising PharmacoTherapy In the multimorbid elderly in primary CAre; SAE, serious adverse event.

the investigators will calculate how many omissions there are.

The MAI and the AOU will be assessed and calculated for baseline, the follow-up one at 6 months and the follow-up two at 12 months. For at least 32 cases (10% of the targeted sample size), two blinded investigators will conduct a blinded independent double assessment of the MAI and the AOU to check inter-rater reliability. The investigators will use information about diagnoses (coded in ICPC-2) and medication intake from the FIRE database for these assessments.

### Assessment of secondary outcomes

The following secondary outcomes are assessed at baseline, 6 and 12 months (data source in brackets)

► Degree of polypharmacy; that is, the number of regular long-term medications patients take (FIRE database).

► Degree of overprescribing, measured by the MAI (assessment of FIRE data done by study team).

► Degree of underprescribing, measured by the AOU (assessment of FIRE data done by study team).

► Number of falls and fractures in the last six months (patient/relative phone call).

► Quality of life, based on the five-level version of the European Quality of Life-5 Dimensions questionnaire, including pain/discomfort[44 45] (patient/relative phone call).

► Amount of formal care received in the last six months: number and length of stay of planned and unplanned hospitalisations; visits to the emergency room without inpatient hospitalisation; GP visits; medical specialist visits (differentiated by specialty); hospital outpatient visits; inpatient stays; length of stay at rehabilitation facilities; physiotherapist and other allied therapist visits; nursing home admissions (in patients who were living in the community at baseline); length of stay in nursing homes; and, number of home nursing visits (patient/relative phone call, except for number of GP visits which comes from FIRE database).

► Amount of informal care received in the last six months: unpaid care by, for example, family members, relatives, friends (patient/relative phone call).

**Table 2** Criteria of the Medication Appropriateness Index including weights

| | Item | Weight |
|---|---|---|
| 1 | Is there an indication for the drug? | 3 |
| 2 | Is the medication effective for the condition? | 3 |
| 3 | Is the dosage correct? | 2 |
| 4 | Are the directions correct? | 2 |
| 5 | Are the directions practical? | 1 |
| 6 | Are there clinically significant drug–drug interactions? | 2 |
| 7 | Are there clinically significant drug–disease/condition interactions? | 2 |
| 8 | Is there unnecessary duplication with other drug(s)? | 1 |
| 9 | Is the duration of therapy acceptable? | 1 |

Item 10 'Is this drug the least expensive alternative compared to others of equal utility?' has been excluded.[40]

► Survival (patient/relative phone call or reported by GP).
► QALYs accrued in one year,[46] measured as a function of the length and the quality of life (calculated based on data from patient/relative phone call).
► Direct medical costs accrued in one year, measured by combining formal and informal healthcare resource use observed in the trial (patient/relative phone call), including the time GPs spend on the intervention (reported by GP) and software costs (literature search), and Swiss unit costs from sources external to the trial (literature search).
► Cost-effectiveness of the intervention, calculated by combining clinical data (FIRE database), quality of life data (patient/relative phone call) and healthcare use data collected in the trial (patient/relative phone call).

The following outcome will be assessed after the intervention
► Percentage of STRIPA recommendations accepted and rejected by GPs (reported by GPs, cross-verified with STRIPA reports).

The following secondary outcome is assessed at baseline only:
► Patients' willingness to have medications deprescribed, measured with the validated 'revised Patient Attitudes Towards Deprescribing' questionnaire[47 48] (patient/relative phone call).

Safety outcomes include adverse events, serious adverse events and device deficiencies.

## Study duration

GPs can recruit the patients over a period of at least six months, so they can integrate recruitment into their daily practice. Recruited patients are followed up for one year.

## Sample size

Sample size calculation is based on testing the two co-primary outcomes for superiority and uses the Bonferroni-approach to account for multiple testing. Based on trial results published by Gallagher *et al* in 2011, the investigators assumed that 35% of patients in the control group and 60% of patients in the intervention group will improve their total MAI score (at least one less point) and that 10% of patients in the control group and 30% of patients in the intervention group will have a better AOU score (at least one fewer prescribing omission).[49] Intracluster correlations (ICC) of 0.01–0.05 are typically found for binary outcomes in elderly individuals.[50] The investigators conservatively assumed an ICC of 0.05 to calculate the sample size. The investigators fixed the type I error at a Bonferroni-corrected two-sided alpha level of 0.025.

Based on a two-sample comparison of proportions and a prespecified number of clusters of about 40 (about 20 per arm), seven patients per cluster are required to detect a difference of 25% between the two groups in the proportion of improvement in the MAI, with a power of 90%. The number of 40 GPs was selected arbitrarily for feasibility reasons. Using the same assumption for the AOU, the investigators found they also need seven patients per cluster to detect a difference of 20%. The investigators thus need a sample of 280 patients (140 per arm) to provide 81% power to detect a significant improvement in both the MAI score and the AOU. To account for attrition from dropout or death (15% estimated), the number of patients per cluster was increased to eight (max. ten), so the final sample size should be about 320 patients (160 per arm). The investigators will closely support GPs to help them reach the target sample size.

## Statistical analysis

In case data on outcomes are incomplete, the investigators will use multiple imputation to replace missing values, taking data clustering into account. No interim analyses are planned.

There are two co-primary outcomes: improvement in the MAI at 12 months, defined as decrease of at least one point, and improvement in the AOU at 12 months, defined as at least one less prescribing omission. Both outcomes will be tested separately; success is indicated by the significance of at least one of the two tests.

For the co-primary outcomes, the investigators will present and compare the proportion of patients whose MAI and AOU score improved in the control and intervention group. The relative difference between groups will be determined in a mixed-effects logistic model with a random intercept at the GP level to account for clustering. The effect measure for the primary outcomes will be odds ratios (OR). The relative difference will be presented as OR with a 95% CI. The primary analysis will be an intention-to-treat analysis. In a per-protocol

analysis, the investigators will only evaluate patients who adhered to the protocol and exclude patients who violate any inclusion or exclusion criteria.

Secondary binary outcomes will be evaluated like the primary outcomes. Secondary continuous outcomes will be analysed using random-effects linear regression with a random intercept at the GP level. Models will also be adjusted for the baseline value as a covariate. Secondary count outcomes will be analysed using random-effects Poisson regression with a random intercept at the GP level.

Health economic analyses will assess (i) resource use and cost differences between the trial arms, (ii) differences in quality-adjusted lifetime between the trial arms, expressed as QALYs, and (iii) a comparison of cost-effectiveness between intervention and standard care.

Because cluster-randomisation may not balance characteristics between groups to match individual-level randomisation, the investigators will adjust each model for patient-level and physician-level variables to account for case-mix differences between groups and potential recruitment bias in a sensitivity analysis. They will also account for the correlated nature of data among GPs by using multilevel mixed-effects models. Unadjusted models will be provided for information only.

### Patient and public involvement

Multimorbid, older patients with polypharmacy are represented in the independent Safety and Data Monitoring Board of the OPTICA trial. Throughout the trial, the Safety and Data Monitoring Board, which will consist of one GP, one statistician and one multimorbid, older layperson with polypharmacy, will meet regularly to discuss safety and data management issues. Patients are not actively involved in recruiting study participants, but play a key role in shared decision-making during the intervention and sham intervention. The investigators have created a priority list of questionnaire components to reduce the burden of the intervention on very old and sick study participants by reducing the duration of the follow-up calls where necessary. At the end of the study, the investigators will disseminate the results to study participants in a letter.

### DISCUSSION

This protocol paper highlights the features of the OPTICA trial, the first RCT in primary care to test an intervention based on the STRIPA clinical decision support tool, which helps GPs customise medication reviews and optimise polypharmacy in older multimorbid patients.

This clinical trial compares the effect of a structured medication review on medication appropriateness in a Swiss primary care setting to a sham intervention. Systematic medication review may facilitate shared decision-making and improve medication appropriateness, especially for GPs who treat complex multimorbid patients with polypharmacy. It may also improve patients'

quality of life and health economic outcomes. This trial will add to the literature, as it examines in a real life setting a software-based intervention, which implements the STOPP/START criteria, based on data from electronic medical records. If successful, this study will demonstrate the usefulness of an electronic database, with coded data collected routinely in primary care, to be used in a clinical decision support tool. Additionally, it focuses on multimorbid patients who are often excluded from trials.

OPTICA is subject to the following limitations

► Despite taking precautions to avoid and reduce selection bias (cluster randomisation to avoid learning effect, screening lists for patient recruitment, randomisation of GPs after patient recruitment), selection bias is still possible because of the study design. Since GPs can recruit patients outside of the screening list for practical reasons, not all patients are recruited randomly. Patients who are more engaged and/or more likely to adhere to advice may be more likely to be enrolled in the trial, which might decrease the representativeness of the study population and the generalisability of the results. The investigators will use data from FIRE to compare the characteristics of study participants with those of non-participants.

► The investigators chose a usual care sham intervention in the control group to improve patient blinding, but this design does not eliminate the risk GPs in the control group will be contaminated by the thematic of the trial (risk of deviating from their previous routine prescribing practices). In addition, this sham intervention might lead GPs to suggest medication changes that could improve the appropriateness of patients' medications.

► Outcome assessment is based on self-reported data from patients and relatives and on data from FIRE, so some events may be missed. To ensure FIRE data is as complete as possible, GPs must code their patient's medication intake and their diagnoses correctly. Diagnoses require ICPC-2 codes. GPs were instructed, in face-to-face meetings, about how to code and update data in their EMR for patients in this trial.[26]

► Like all interventions that take an eHealth approach (STRIPA intervention in this case), people have different perceptions of what is user-friendly. New procedures may seem complicated in comparison to usual care. The investigators tried to simplify the process by (i) using EMR data so that GPs do not have to enter all the data themselves, but only update them if needed; and (ii) giving detailed instructions for using STRIPA in writing and in an online video training.

► Unrestricted randomisation designs, such as the block randomisation used in this trial, are more likely to result in imbalance of factors by chance.[51]

► Another limitation of the study is the restriction to self-control during the intervention, as it is a mono-professional intervention, and that the scope

of the intervention is limited to the use of the software-based CDSS. However, due to the structure of the Swiss primary healthcare setting this design is feasible in a real life setting, whereas multiprofessional interventions would be difficult to organise.

▶ The primary outcome of this trial is not directly patient-relevant. However, directly patient-relevant outcomes, such as quality of life, figure among the secondary outcomes of this trial.

It may be difficult to follow-up multimorbid, older patients, who are often excluded from trials.[52] The investigators believe that the task is worth the effort because trial results need to be generalisable to exactly this population. To reduce the duration of the phone calls with particularly weak study participants only the core elements of the study questionnaire will be used, if necessary, based on a predefined priority list.

The OPTICA trial has the following strengths:

▶ The OPTICA trial is the first RCT to examine the effect of STRIPA on medication appropriateness in older, multimorbid patients with polypharmacy in a primary care setting.

▶ OPTICA is the first RCT to test the use of software-based structured medication reviews in Swiss primary care.

▶ OPTICA demonstrates the usefulness of coding and linking data from EMR, and re-using this data to evaluate primary care interventions in a randomised controlled trial setting.

▶ In the OPTICA trial, the investigators do not exclude patients with cognitive impairment if a relative gives their informed consent, because patients with cognitive impairment are especially prone to polypharmacy, yet they have been excluded from many other trials.

## CONCLUSION

The OPTICA trial will compare the effect of the systematic medication review that uses STRIPA, including shared decision-making, to usual care (sham intervention) with the goal of improving medication appropriateness. The investigators expect the intervention to improve the quality of life and health status of a rapidly ageing population with increasing multimorbidity and polypharmacy. The study results will inform other studies and interventions designed to optimise medication use by integrating a CDSS with electronic medical records.

## Current status of the OPTICA trial

The patient recruitment in the OPTICA trial began in December 2018. By early July 2019, 278 patients (about 85% of the target sample size) have been recruited. The investigators have randomised 31 out of about 40 GPs. Last patient out is expected in the second half of 2020.

## Ethics

All participant data will be handled according to the principles of the Declaration of Helsinki.[53] The OPTICA trial complies with all applicable standards of the guideline for Good Clinical Practice of the International Conference on Harmonisation (ICH-GCP).[54]

Data management, monitoring, safety reporting and audits meet the requirements of the Swiss law. The investigators uphold the principle of patients' right to privacy and comply with applicable privacy laws. Confidentiality and anonymity of the patients shall be guaranteed when the data is presented at scientific meetings or published in scientific journals. Only selected study team members will have access to the final trial dataset.

Risks, including human failure and software malfunction, cannot be excluded. But STRIPA only makes prescription recommendations to GPs in the intervention group. Participating GPs are experienced (mean age of experience as GP: 16 years), and will take the final decision about whether to accept the recommendations and to present them to the patient. Patients thus are not exposed to more risk than they would be in standard care. This clinical trial entails minimal risk for participants and the benefit-risk ratio is positive. *Basler Versicherungen* will provide insurance and cover eventual damages.

Participating GPs have signed a non-disclosure agreement.

## Dissemination

OPTICA embraces an open access policy and will vigorously disseminate all resulting data, study results and publications. The investigators closely collaborate with the National Research Programme 74 (NRP74) 'Smarter Health Care' to optimise dissemination of the study results to the public.

**Author affiliations**
[1]Institute of Primary Health Care (BIHAM), University of Bern, Bern, Switzerland
[2]Department of General Internal Medicine, Inselspital, Bern University Hospital, University of Bern, Bern, Switzerland
[3]Institute of Primary Care, University of Zurich, Zurich, Switzerland
[4]Department of Information and Computing Sciences, Utrecht University, Utrecht, The Netherlands
[5]Department of Communication and Cognition, Tilburg University, Tilburg, The Netherlands
[6]Department of Geriatric Medicine, University Medical Centre Utrecht, Utrecht, The Netherlands
[7]Department of Clinical Pharmacy, University Medical Centre Utrecht, Utrecht, The Netherlands
[8]Department of Public Health and Primary Care, Leiden University Medical Center, Leiden, The Netherlands
[9]Quality Use of Medicines and Pharmacy Research Centre, School of Pharmacy and Medical Sciences, Division of Health Sciences, University of South Australia, Adelaide, South Australia, Australia
[10]Geriatric Medicine Research, Faculty of Medicine, Dalhousie University and Nova Scotia Health Authority, Halifax, Nova Scotia, Canada
[11]Canada College of Pharmacy, Faculty of Health, Dalhousie University, Halifax, Nova Scotia, Canada
[12]Hausarzt Praxis Städtli Aarburg, Aarburg, Switzerland
[13]Berufsverband der Haus- und Kinderärztinnen Schweiz, Bern, Switzerland
[14]CTU Bern, University of Bern, Bern, Switzerland
[15]Institute of Pharmaceutical Medicine (ECPM), University of Basel, Basel, Switzerland
[16]Epidemiology, Biostatistics and Prevention Institute, University of Zurich, Zurich, Switzerland

**Acknowledgements** The entire OPTICA study team would like to acknowledge the enormous help provided by all general practitioners participating in the OPTICA trial and thank all the participating GPs for their efforts to recruit older multimorbid patients with polypharmacy. The investigators also thank Kali Tal for her editorial suggestions.

**Contributors** Authorship eligibility is based on the four ICMJE authorship criteria. Study concept and design: ST, MS, MS, KTJ, HB, RM, NR, SS. Drafting of the manuscript: KTJ, SS. Critical contribution to the study protocol and revision of the manuscript for important intellectual content: SM, CF, ALL, FL, ZR, NS, RM, BTGMS, CJAH, MCM, LE, CS, MF, HB, ER, PMB, ST, MS, MS, NR. Statistical analysis: ST. Obtained funding: NR, SS. Administrative, technical, or material support: NS. Supervision: SS.

**Funding** This work is supported by the Swiss National Science Foundation, within the framework of the National Research Programme 74 (NRP74) under contract number 407440_167465 (to SS and NR).

**Competing interests** None declared.

**Patient consent for publication** Not required.

**Ethics approval** The ethics committee of the canton of Bern (Switzerland) and the Swiss regulatory authority (Swissmedic) approved the study protocol and other documentation on study conduct (BASEC ID: 2018–00914). The ethics committee and Swissmedic will receive annual safety reports and information about study stops/end and protocol amendments, as per local requirements.

**Provenance and peer review** Not commissioned; externally peer reviewed.

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
