## [Reviewer comments · BMJ Open]

ARTICLE DETAILS

TITLE (PROVISIONAL)	'Optimizing Pharmacotherapy In the Multimorbid Elderly in Primary Care' (OPTICA) to improve medication appropriateness: study protocol of a cluster randomised controlled trial
AUTHORS	Jungo, Katharina; Rozsnyai, Zsofia; Mantelli, Sophie; Floriani, Carmen; Löwe, Axel; Lindemann, Fanny; Schwab, Nathalie; Meier, Rahel; Elloumi, Lamia; Huibers, Corlina; Salleveld, Bastiaan; Meulendijk, Michiel; Reeve, Emily; Feller, Martin; Schneider, Claudio; Bhend, Heinz; Bürki, Pius; Trelle, S; Spruit, Marco; Schwenkglens, Matthias; Rodondi, Nicolas; Streit, Sven

VERSION 1 - REVIEW

REVIEWER	Barbara Clyne Royal College of Surgeons in Ireland
REVIEW RETURNED	29-May-2019

GENERAL COMMENTS	Many thanks for asking me to review this paper. The authors are to be commended on a well described and conceptualised randomised controlled trial protocol in a complex area. Overall, this protocol is well described and the authors have paid good attention to reducing bias as far as possible within their design. Some further minor clarifications would improve the clarity of the paper overall 1. Title: 'design of' could be replace with 'protocol for' as design makes it seem more like a description of the development of the intervention2. I would expect there to be more reference to important research in the fields of multimorbidity and polypharmacy research, for example, recent Cochrane reviews on interventions:a. Rankin A, Cadogan CA, Patterson SM, Kerse N, Cardwell CR, Bradley MC, Ryan C, Hughes C. Interventions to improve the appropriate use of polypharmacy for older people. Cochrane Database of Systematic Reviews 2018, Issue 9. Art. No.: CD008165. DOI: 10.1002/14651858.CD008165.pub4.b. Smith SM, Wallace E, O'Dowd T, Fortin M. Interventions for improving outcomes in patients with multimorbidity in primary care and community settings. Cochrane Database of Systematic Reviews 2016, Issue 3. Art. No.: CD006560. DOI: 10.1002/14651858.CD006560.pub3.c. Other RCTs which have used informatics to facilitate medication reviews with older adults such as Dreischulte T, Donnan P, Grant A, Hapca A, McCowan C, Guthrie B. Safer Prescribing-A Trial of
---

	Education, Informatics, and Financial Incentives. N Engl J Med. 2016 Mar 17;374(11):1053-64. 3. Such previous work would make me question if it is truly legitimate to make the claim that this is the first RCT in this area 4. Randomisation: In cluster RCTs, simple randomisation is more likely to result in imbalance in factors by chance, because there are often relatively few clusters. Restricted designs are therefore more appropriate. This is a limitation of the design that should be acknowledged - Sandra Eldridge Sally Kerry. A Practical Guide to Cluster Randomised Trials in Health Services Research 5. Recruitment: GP recruitment commenced in 2017, what has been happening in the interim? Why the delay in publishing the protocol until now? 6. Secondary outcomes: from the list provided on under 'assessment of secondary outcomes', the data source for some of these outcomes is not clear. For example, where will number of falls be collected from? 7. Sample size: a. 40 GP practices were recruited. What is the justification for 40? b. "The investigators assumed that 35% of patients in the control group and 60% of patients in the intervention group will improve their total Medication Appropriateness Index (MAI) score – based on what?
--	---

REVIEWER	Olaf Rose impac2t, Germany
REVIEW RETURNED	03-Jun-2019

GENERAL COMMENTS	Dear authors of the OPTICA study, Thank you for providing this interesting study design to us. It was prepared with great accuracy; however, I would recommend to describe the calculation of the MAI more detailed. Please make sure not to mix overprescribing with appropriateness of medication, a medication review (and the MAI) can assess more than just over- and underprescribing, it can detect the whole range of drug-related problems. Hope my revisions will be of help to you in publishing an even better paper. P4L4 bullet 1: several studies of a systematic structured MR on MAI have been conducted before (Köberlein-Neu et al. Interprofessional Medication Management in Patients with Multiple Morbidities, PMID: 27890050), please check literature and rephrase P4L8 bullet 2: how can you present results before the study is finished/in a study design publication? P4L12: sentence difficult to understand, please rephrase L35 inappropriate polypharmacy surely is more than just the prescription of too many drugs, it can be vice versa as well (indication but no drug) but in most cases is misprescribing (wrong drug/dose for the indication). I recommend the use of an exact definition, change denote to 'as an example can denote' or delete the sentence. Whole paragraph: consider to change to a comprehensive definition rather than placing some examples. P5 L22
--

	patients' willingness to deprescribe, overprescribing and underprescribing. Please amend to 'patients' acceptance of physician`s changes to deprescribing, overprescribing and underprescribing' (if I got you right). P9 patient selection: please state more clearly that patients are not recruited randomly (as the GP can add patients to the list). This should also be explained in the first box of table 1. P10L56-60 the Mai assesses among other items, overprescribing (please make sure that the MAI is not just a tool to assess overprescribing but thoroughly the quality of the medication). the AOU assesses underprescribing (not over) P11: please state more precisely, how you are going to calculate the MAI. Do you dichotomize (only inappropriate as 1, appropriate and marginally appropriate =0)? Do you use the weighted score according to Samsa et al. (you do if you reach 18 per drug, so please state and cite)? Do you sum up afterwards the score per drug, etc.pp. Please describe the MAI more detailed to be comparable to other studies. Do you calculate the MAI each time you assess the patient? Is MAI calculation done by 1 or 2 researchers? P11 L59: same as before: willingness to accept deprescribing P14: another limitation of the study is the restriction to self-control (a MR is usually interprofessional) and the limitation of the intervention to the decision support system (a comprehensive MR could be based on a more comprehensive assessment (handling of drugs, guideline adherence, etc.). This is especially true, as the MAI covers more than under- and overprescribing, it detects misprescribing as well. Discussion should include your project in the light of other MR projects performed so far. What is different (monoprofessional, decision-support system based), what is new if compared to other landmark studies in MR?
--	---

REVIEWER	Torgeir Bruun Wyller Dept. of Geriatric Medicine Institute of Clinical Medicine University of Oslo, Norway
REVIEW RETURNED	04-Jun-2019

GENERAL COMMENTS	The authors present the protocol of a cluster RCT to evaluate the effect of the OPTICA (Optimizing Pharmacotherapy In the Multimorbid Elderly in Primary Care) digital prescription tool. They have two co-primary endpoints: scores on the Medication Appropriateness Index (MAI) and on the Assessment of Underutilization (AOU). They also describe several secondary endpoints, among them are patient relevant outcomes like quality of life (QoL) as assessed with the EQ-5D instrument, number of falls and fractures, and number of hospitalizations. The authors plan to include 320 patients in 40 clusters, and the recruitment is ongoing.
---

	Polypharmacy and drug related harm in frail elderly patients is a major concern. Methods for optimizing prescriptions, focusing upon total drug regimens, not only on single drugs, are severely under-studied. High quality studies that can provide evidence based methods for handling of these issues are therefore highly welcome, and clearly have the potential to change future medical practice. I very much appreciate the authors attempt to test the intervention in a patient group that is as unselected as possible (in principle only selected based on age and degree of polypharmacy). It is also very laudable that e-health tools are subject to stringent trials to test their clinical effectiveness. One concern with this project is that MAI and AOU are not necessarily patient relevant outcomes. They are theoretical measures, reflecting what experts think are appropriate and inappropriate use of drugs. As such, they reflect many of the same ideas as the STOPP and START criteria, on which the OPTICA tools is based. Thus, I see some risks of a circular argumentation here. It is likely that a tool that is based on expert opinion on medication appropriateness may influence prescription in a way that is in accordance with just these ideas. It is a very good thing that the MAI and AOU scores are supplemented by secondary outcomes that are directly patient relevant. The protocol article can be improved, however, if the authors spend some lines in the Discussion to reflect over strengths and weaknesses with the chosen primary outcomes. Assessment of outcomes will be based on telephone interview. What do we know about the validity of for instance fall reports and reports of QoL, obtained by telephone interview from respondents that may be cognitively impaired? In my opinion this should also be critically discussed. The authors plan to report quality adjusted living years (QALYs). This is a composite measure, comprising survival and an estimate of QoL. It is generally recommended that when composite endpoints are used in clinical trials, the elements they comprise should also be reported separately. Accordingly, besides the EQ-5D score, the authors should also report survival. If QALY is affected by the intervention, the reader should be able to assess whether the change was driven by survival, by the QoL estimate, or by both. In conclusion, this trial has the potential to become important, and I look forward to see the results.
--	---

VERSION 1 – AUTHOR RESPONSE

Reviewer(s)' Comments to Author:

Reviewer: 1

Reviewer Name: Barbara Clyne

Institution and Country: Royal College of Surgeons in Ireland Please state any competing interests or state 'None declared': None declared

Many thanks for asking me to review this paper. The authors are to be commended on a well described and conceptualised randomised controlled trial protocol in a complex area. Overall, this protocol is well described and the authors have paid good attention to reducing bias as far as possible within their design. Some further minor clarifications would improve the clarity of the paper overall.

Response: We thank the reviewer for the positive comments on our paper. All points mentioned are addressed point-by-point below.

1. Title: 'design of' could be replaced with 'protocol for' as design makes it seem more like a description of the development of the intervention

Response: The title has been adapted accordingly, so that it is now clearly stated that it is a protocol paper (page 1, line 1-3).

2. I would expect there to be more reference to important research in the fields of multimorbidity and polypharmacy research, for example, recent Cochrane reviews on interventions:

a. Rankin A, Cadogan CA, Patterson SM, Kerse N, Cardwell CR, Bradley MC, Ryan C, Hughes C. Interventions to improve the appropriate use of polypharmacy for older people. Cochrane Database of Systematic Reviews 2018, Issue 9. Art. No.: CD008165. DOI: 10.1002/14651858.CD008165.pub4.

b. Smith SM, Wallace E, O'Dowd T, Fortin M. Interventions for improving outcomes in patients with multimorbidity in primary care and community settings. Cochrane Database of Systematic Reviews 2016, Issue 3. Art. No.: CD006560. DOI: 10.1002/14651858.CD006560.pub3.

c. Other RCTs which have used informatics to facilitate medication reviews with older adults such as Dreischulte T, Donnan P, Grant A, Hapca A, McCowan C, Guthrie B. Safer Prescribing-A Trial of Education, Informatics, and Financial Incentives. N Engl J Med. 2016 Mar 17;374(11):1053-64.

Response: We have incorporated these references in the introduction of our protocol paper on page 3 and 4. (Detailed information: page 3, line 32 to page 4, line 4 / page 4, line 8-15 / page 4, line 18-20)

3. Such previous work would make me question if it is truly legitimate to make the claim that this is the first RCT in this area

Response: We thank the reviewer for highlighting this. This claim has been replaced by the two following statements on page 2, line 5-10:

- "The OPTICA trial is the first randomized controlled trial to examine the effect of the 'Systematic Tool to Reduce Inappropriate Prescribing'-Assistant (STRIPA), a software-assisted clinical decision support tool, on medication appropriateness in older, multimorbid patients with polypharmacy in a primary care setting", as the STRIPA has only been tested in a European multicentre trial in the hospital setting (the OPERAM trial)

-“OPTICA is the first randomized controlled trial to test the use of a software-based structured medication review in Swiss primary care”, since there has not been any previous trial using software-based structured medication review in Switzerland’s primary health care setting

4. Randomisation: In cluster RCTs, simple randomisation is more likely to result in imbalance in factors by chance, because there are often relatively few clusters. Restricted designs are therefore more appropriate. This is a limitation of the design that should be acknowledged - Sandra Eldridge Sally Kerry. A Practical Guide to Cluster Randomised Trials in Health Services Research

Response: We thank the reviewer for this comment. In the OPTICA trial, we are not using simple randomization, but block randomization to make sure that we have a similar number of study participants in both trial arms. We apologize for this confusion and we have adapted the text accordingly. As mentioned, block randomization is less efficient in avoiding imbalances in factors than restricted randomization. We have thus added this limitation to our limitation section on page 15, line 30-31: “Unrestricted randomisation designs, such as the block randomization used in this trial, are more likely to result in imbalance of factors by chance”

5. Recruitment: GP recruitment commenced in 2017, what has been happening in the interim?

Why the delay in publishing the protocol until now?

Response:

We thank the reviewer for this comment and we are happy to clarify this:

On the one hand, our study team started the recruitment of GPs in 2017 already, as we were not able to anticipate how long it would take to recruit 40 motivated GPs for this trial. Previous experience concerning RCTs in the primary health care setting had shown that the recruitment of GPs may be major obstacle to clinical research in this setting (Huibers et al. 2004, Yallop et al. 2006), which is why we developed and started our recruitment strategy early on in the project. To recruit GPs we organized practice visits to all interested GPs, during which we explained the study in detail. We were able to do this before receiving ethical approval as GPs are not recruited in the same sense that study participants are, since they are sub-investigators in this trial and not study participants. We clarified this in the manuscript on page 5, line 19-20.

On the other hand, we only received conditional approval of the trial by the competent ethics committee in November 2018. The conditions were: to send confirmation that all GPs read the for them relevant parts of the study protocol, to correct a mistake in patient information form, to send confirmation of GPs having done the online GCP tutorial, and to submit the finalized monitoring report. Only then, we were able to start the study.

A first draft of the protocol paper had already been prepared in autumn 2018. Due to the need for minor final adaptations because of the conditional approval when the study started, we waited with publishing the protocol paper until everything was set in order to avoid that the protocol paper might contain outdated information.

6. Secondary outcomes: from the list provided on under ‘assessment of secondary outcomes’, the data source for some of these outcomes is not clear. For example, where will number of falls be collected from?

Response: For every secondary outcome, we have now specified the data source. Please refer to page 11, line 19 to page 12, line 30 for details.

7. Sample size:

- a. 40 GP practices were recruited. What is the justification for 40?

Response: For feasibility reasons, the number of 40 clusters was pre-specified when doing the sample calculation. The decision that 40 clusters will be needed for this trial was made given that at the time, only about 150 GPs were part of the FIRE project. Recruiting 40 GPs meant that one quarter of these GPs had to be recruited for the OPTICA trial. We have specified this on page 13, line 9-10.

The FIRE project is a research project under the direction of the Institute for Family Medicine at the University of Zurich (IHAMZ). FIRE stands for "Family medicine ICPC Research using Electronic medical records". The acronym summarizes the objectives of the project: The development of a database for research in family medicine based on routine medical data from electronic medical records, whereby diagnoses are classified according to the International Classification of Primary Care (ICPC).

- b. "The investigators assumed that 35% of patients in the control group and 60% of patients in the intervention group will improve their total Medication Appropriateness Index (MAI) score – based on what?"

Response: We thank the reviewer for highlighting that a reference was missing here. We had based this assumption on a trial published by Gallagher et al. in 2011. We have added this information in the manuscript on page 13, line 2-3.

Reviewer: 2

Reviewer Name: Olaf Rose

Institution and Country: impac2t, Germany Please state any competing interests or state 'None declared': None declared

Dear authors of the OPTICA study, Thank you for providing this interesting study design to us. It was prepared with great accuracy; however, I would recommend to describe the calculation of the MAI more detailed. Please make sure not to mix overprescribing with appropriateness of medication, a medication review (and the MAI) can assess more than just over- and underprescribing, it can detect the whole range of drug-related problems. Hope my revisions will be of help to you in publishing an even better paper.

Response: We thank the reviewer for the positive feedback on our paper. The comments were very helpful.

We added a detailed description of the Medication Appropriateness Index to the text on page 10 and the criteria of the MAI, including the weights assigned, in table 2 (page 11, line 9). In addition, we

have clarified that the Medication Appropriateness Index can assess more than just overprescribing. Overprescribing is just one way of looking at medication appropriateness (page 10, line 12-14).

P4L4 bullet 1: several studies of a systematic structured MR on MAI have been conducted before (Köberlein-Neu et al. Interprofessional Medication Management in Patients with Multiple Morbidities, PMID: 27890050), please check literature and rephrase

Response: We thank the reviewer for highlighting the issue with this statement. The statement was replaced with the following two sentences that also figure in the strengths section of this paper (page 3, line 5-9):

-“The OPTICA trial is the first randomized controlled trial to examine the effect of 'Systematic Tool to Reduce Inappropriate Prescribing'-Assistant (STRIPA) a software-assisted clinical decision support tool, on medication appropriateness in older, multimorbid patients with polypharmacy in a primary care setting.”

-“OPTICA is the first randomized controlled trial to test the use of software-based structured medication review in Swiss primary care.”

P4L8 bullet 2: how can you present results before the study is finished/in a study design publication?

Response: Thank you for this comment. This statement is not about the study results of the OPTICA trial, but instead it is about the application of the use of coded data from electronic medical records in a randomized controlled trial. To be clearer we have rephrased the statement to: “The OPTICA trial demonstrates how linked and coded data from electronic medical records can be used to evaluate primary care interventions in a randomized controlled trial setting” (page 3, line 10-11).

P4L12: sentence difficult to understand, please rephrase

Response: The sentence has been split up into two separate sentences (page 4, line 24-35):

“The primary aim of the cluster-randomised ‘Optimizing PharmacoTherapy In the Multimorbid Elderly in Primary Care’ (OPTICA) trial is to test whether the use of a systematic, software-assisted medication review intervention leads to a more appropriate use of medications than a usual care sham intervention. This primary outcome will be measured by the Medication Appropriateness Index (MAI) and the Assessment of Underutilization (AOU). Secondary outcomes are the degree of polypharmacy, the degree of overprescribing, the degree of underprescribing, the number of falls and fractures, quality of life, the amount of formal and informal care received by patients, survival, patients’ quality adjusted life years, patients’ medical costs, cost-effectiveness of the intervention, percentage of recommendations accepted and rejected by GPs, and patients’ willingness to have medications deprescribed.”

L35 inappropriate polypharmacy surely is more than just the prescription of too many drugs, it can be vice versa as well (indication but no drug) but in most cases is misprescribing (wrong drug/dose for the indication). I recommend the use of an exact definition, change denote to ‘as an example can denote’ or delete the sentence.

Whole paragraph: consider to change to a comprehensive definition rather than placing some examples.

Response: As suggested, the sentence about inappropriate polypharmacy has been deleted. We rewrote parts of the first paragraphs to make the content of the introduction more coherent and comprehensible (page 3, line 29-30 / page 3, line 32-33). The added sentences are highlighted in yellow.

P5 L22 - patients' willingness to deprescribe, overprescribing and underprescribing.

Please amend to 'patients' acceptance of physician's changes to deprescribing, overprescribing and underprescribing' (if I got you right).

Response: The secondary outcome "patients' willingness to deprescribe" is assessed by a validated questionnaire called "revised Patient Attitudes Towards Deprescribing" (rPATD), developed by Reeve et al. in Australia, at baseline. It is thus not related to any specific medication changes suggested by the general practitioners, but it is rather a measure of a patients' general willingness to have medications deprescribed, or in other words, patients' attitudes towards deprescribing. Because of the officially established wording of the rPATD questionnaire, we have decided to keep the wording in line with the original literature published on the rPATD, which is why we have changed it to "willingness to have medications deprescribed" (page 2, line 29 / page 4, line 35 / page 12, line 28).

In addition, we have changed the order in which the secondary outcomes are listed to make this list easier to read. The degree of over- and underprescribing now figure after the degree of polypharmacy, so that all directly medication-related outcomes come one after another (page 2, line 25 / page 4, line 25 / page 11, line 31-32).

P9 patient selection: please state more clearly that patients are not recruited randomly (as the GP can add patients to the list). This should also be explained in the first box of table 1.

Response: We thank the reviewer for emphasizing this shortcoming. We have added this statement to the limitations section of the paper (page 15, line 7-9): "Since GPs can recruit patients outside of the screening list for practical reasons, not all patients are recruited randomly". In addition, the sentence "However, since GPs can recruit patients outside this list, not all study participants are recruited randomly." has been added to the first box of table 1 on page 9, starting on line 19.

P10L56-60 the MAI assesses among other items, overprescribing (please make sure that the MAI is not just a tool to assess overprescribing but thoroughly the quality of the medication).

Response: We thank the reviewer for this comment. We have rephrased this section: "The primary endpoint is medication appropriateness, measured jointly by the Medication Appropriateness Index (MAI) and the Assessment of Underutilization (AOU), in each study group. While the AOU assesses underprescribing, the MAI is a tool to assess different elements of medication prescribing (e.g. overprescribing, drug-drug interactions, etc.)" (page 10, line 11-14).

the AOU assesses underprescribing (not over)

Response: We have corrected this error.

P11: please state more precisely, how you are going to calculate the MAI.

Response: Thank you for highlighting this shortcoming. More information about the calculation of the Medication Appropriateness Index has been added to the manuscript (page 10, line 16 up to page 11, line 5): "The investigators will use the weighted 10-item version of the MAI developed by Samsa et al.⁴⁰ However, due to the rapidly changing drug prices the MAI item on costeffectiveness will not be included. The criteria of the MAI, including corresponding weights, can be found in table 2. Using clinical data and the predefined operational definitions for each item, the assessor rates each medication on a three-point scale ranging from A = appropriate, B = marginally appropriate, C = inappropriate. Each "inappropriate" rating will receive the respective weight from Table 2, while the weight of ratings "appropriate" and "marginally appropriate" will be 0. Thus, the score for each medication ranges from 0 to 17 (as cost-effectiveness criterion will not be included). A higher score indicates a greater degree of medication inappropriateness. The investigators will calculate the score for each medication and then calculate the summated score for each patient, by summing up the scores for each medication."

Do you dichotomize (only inappropriate as 1, appropriate and marginally appropriate =0)?

Response: Each "inappropriate" rating will receive the respective weight from Table 2, while the weight of ratings "appropriate" and "marginally appropriate" is 0. This information has been added on page 10, line 21-23 up to page 11, line 1.

Do you use the weighted score according to Samsa et al. (you do if you reach 18 per drug, so please state and cite)?

Response: We will indeed use the weighted score according to Samsa et al. However, due to feasibility issues (i.e. rapidly changing drug prices) we will exclude the item on cost-effectiveness. Due to this, the MAI score for each medication ranges between 0 and 17. This is now clearly stated and referenced in the text on page 10, line 17.

Do you sum up afterwards the score per drug, etc.pp.

Response: We will calculate the score for each medication and then calculate the summated score for each patient, by summing up the scores for each medication that the patient is taking. This information has been added to the text on page 11, line 3-5.

Please describe the MAI more detailed to be comparable to other studies.

Response: We have added more details about how we will use the MAI in this study on page 10 up to page 11. We hope that this will make the trial more comparable to other studies.

Do you calculate the MAI each time you assess the patient?

Response: The MAI will be calculated for baseline, follow-up at 6 months and the follow-up at 12 months. This information has been added to the text on page 11, line 11-12.

Is MAI calculation done by 1 or 2 researchers?

Response: The MAI and AOU assessment will be done by two assessors from our study team (physicians). For at least 32 cases (10% of the targeted sample size), the two investigators will conduct a blinded independent double assessment of the MAI and the AOU to check inter-rater reliability as indicated on page 11, line 12-13.

P11 L59: same as before: willingness to accept deprescribing

Response: As mentioned above (on page 6), the wording has been changed to “patients’ willingness to have medications deprescribed” throughout the manuscript (page 2, line 29 / page 4, line 35 / page 12, line 28).

P14: another limitation of the study is the restriction to self-control (a MR is usually interprofessional) and the limitation of the intervention to the decision support system (a comprehensive MR could be based on a more comprehensive assessment (handling of drugs, guideline adherence, etc.). This is especially true, as the MAI covers more than under- and overprescribing, it detects misprescribing as well.

Response: We thank the reviewer for this comment. These two points have been added to the limitations section of the manuscript: “Another limitation of the study is the restriction to self-control during the intervention, as it is a mono-professional intervention, and that the scope of the intervention is limited to the use of the CDSS. However, due to the structure of the Swiss primary care setting this design is feasible in a real life setting, whereas multi-professional interventions would be difficult to organize” (page 15, line 32-36).

Discussion should include your project in the light of other MR projects performed so far. What is different (monoprofessional, decision-support system based), what is new if compared to other landmark studies in MR?

Response: We thank the reviewer for raising this comment. We have adapted the discussion section accordingly. The following sentences have been added to page 14, line 36 to page 15, line 3: “This trial will add to the literature, as it examines in a real life setting a software-based intervention, which implements the STOPP/START criteria, based on data from electronic medical records. If successful, this study will demonstrate the usefulness of an electronic database, with coded data collected routinely in primary care, to be used in a clinical decision support tool. Additionally, it also focuses on multimorbid patients who are often excluded from trials.”

Reviewer: 3

Reviewer Name: Torgeir Bruun Wyller

Institution and Country: Dept. of Geriatric Medicine Institute of Clinical Medicine University of Oslo, Norway Please state any competing interests or state 'None declared':None declared

The authors present the protocol of a cluster RCT to evaluate the effect of the OPTICA (Optimizing Pharmacotherapy In the Multimorbid Elderly in Primary Care) digital prescription tool. They have two co-primary endpoints: scores on the Medication Appropriateness Index (MAI) and on the Assessment of Underutilization (AOU). They also describe several secondary endpoints, among them are patient relevant outcomes like quality of life (QoL) as assessed with the EQ-5D instrument, number of falls and fractures, and number of hospitalizations. The authors plan to include 320 patients in 40 clusters, and the recruitment is ongoing.

Polypharmacy and drug related harm in frail elderly patients is a major concern. Methods for optimizing prescriptions, focusing upon total drug regimens, not only on single drugs, are severely under-studied. High quality studies that can provide evidence based methods for handling of these issues are therefore highly welcome, and clearly have the potential to change future medical practice. I very much appreciate the authors attempt to test the intervention in a patient group that is as unselected as possible (in principle only selected based on age and degree of polypharmacy). It is also very laudable that e-health tools are subject to stringent trials to test their clinical effectiveness.

One concern with this project is that MAI and AOU are not necessarily patient relevant outcomes. They are theoretical measures, reflecting what experts think are appropriate and inappropriate use of drugs. As such, they reflect many of the same ideas as the STOPP and START criteria, on which the OPTICA tools is based. Thus, I see some risks of a circular argumentation here. It is likely that a tool that is based on expert opinion on medication appropriateness may influence prescription in a way that is in accordance with just these ideas. It is a very good thing that the MAI and AOU scores are supplemented by secondary outcomes that are directly patient relevant. The protocol article can be improved, however, if the authors spend some lines in the Discussion to reflect over strengths and weaknesses with the chosen primary outcomes.

Response: We would like the reviewer for this positive reflection on our paper. We appreciate that it has been highlighted that medication appropriateness, as measured by the MAI and the AOU, is not a patient-centred outcome. Because our team had this in mind when developing the study protocol, patient-relevant secondary outcomes, such as quality of life, were included.

The following limitation has been added on page 16, line 1-3: "The primary outcome of this trial is not directly patient-relevant. However, directly patient-relevant outcomes, such as quality of life, figure among the secondary outcomes of this trial".

Assessment of outcomes will be based on telephone interview. What do we know about the validity of for instance fall reports and reports of QoL, obtained by telephone interview from respondents that may be cognitively impaired? In my opinion this should also be critically discussed.

Response: Patients who their GP considers to be cognitively impaired, must be included through the consent of one of their relatives. In such a case, the telephone interviews are done with the relative using specific questionnaires and not with the patient himself/herself.

However, we are also aware of the fact that self-reported data comes with limitations. For instance, patients (without cognitive impairment) tend to underestimate the number of falls as reported by Mackenzie et al. in 2006 and Hannan et al. in 2010. This is already critically discussed in the following limitation on page 15, line 19-23: "Outcome assessment is based on self-reported data from patients and relatives, and on data from FIRE, so some events may be missed. To ensure FIRE data is as complete as possible, GPs must code their patients' medication intake and their diagnoses correctly."

The authors plan to report quality adjusted living years (QALYs). This is a composite measure, comprising survival and an estimate of QoL. It is generally recommended that when composite endpoints are used in clinical trials, the elements they comprise should also be reported separately. Accordingly, besides the EQ-5D score, the authors should also report survival. If QALY is affected by the intervention, the reader should be able to assess whether the change was driven by survival, by the QoL estimate, or by both.

Response: We thank the reviewer for this suggestion. We have added survival to the list of secondary outcomes so that all elements of our composite outcomes are explicitly covered (page 2, line 26 / page 4, line 33 / page 12, line 12).

In conclusion, this trial has the potential to become important, and I look forward to see the results.

Response: We would like to thank the reviewer for this assessment.

VERSION 2 – REVIEW

REVIEWER	Barbara Clyne Royal College of Surgeons in Ireland
REVIEW RETURNED	30-Jul-2019

GENERAL COMMENTS	The authors have addressed all comments satisfactorily, I would recommend this protocol for publication.
--

REVIEWER	Olaf Rose impac2t, Germany
REVIEW RETURNED	09-Aug-2019

GENERAL COMMENTS	Thank you for providing this update to us reviewers. From my perspective, the paper has improved a lot and is ready for publication.
--

REVIEWER	Torgeir Bruun Wyller Department of Geriatric Medicine, University of Oslo
REVIEW RETURNED	30-Jul-2019

GENERAL COMMENTS	I think the authors have adequately answered the points that were raised regarding the first version of the manuscript.
---